# Parental experiences of a diagnosis of neonatal diabetes and perceptions of newborn screening for glucose: a qualitative study

Maggie Shepherd [1,2] Bridget A Knight,[2] Katherine Laskey,[3]
Timothy James McDonald[2,3]

► Prepublication history and additional materials for this paper is available online. To view these files, please visit the journal online (http://dx.doi.org/10.1136/bmjopen-2020-037312).

[1]Institute of Biomedical and Clinical Science, University of Exeter, Exeter, Devon, UK
[2]NIHR Exeter Clinical Research Facility, Royal Devon and Exeter NHS Foundation Trust, Exeter, Devon, UK
[3]Department of Blood Sciences, Royal Devon and Exeter NHS Foundation Trust, Exeter, UK

**Correspondence to**
Prof Maggie Shepherd;
M.H.Shepherd@exeter.ac.uk

## ABSTRACT

Neonatal diabetes presents <6 months of life but delays in recognition result in presentation with life-threatening hyperglycaemia/diabetic ketoacidosis. Early identification and rapid genetic diagnosis is crucial and ensures correct treatment/management. Adding 'glucose' to newborn bloodspot screening (NBS) could aid prompt detection but requires evidence of parental acceptance.

**Objectives** Increase understanding of parental experience of presentation/recognition of neonatal diabetes and perceptions of glucose testing within NBS.

**Setting** UK families confirmed with a genetic diagnosis of neonatal diabetes, November 2014–2018, were invited to participate.

**Participants** In-depth qualitative interviews were conducted with 10 parents of 14 children. 8 had transient neonatal diabetes: *KCNJ11* (n=5), *ABCC8* (n=1), *6q24* (n=2), 6 had permanent neonatal diabetes: *KCNJ11* (n=4), *INS* (n=1), homozygous *GCK* (n=1).

**Primary and secondary outcome measures** Interviews audio recorded, transcribed and subjected to thematic content analysis.

**Results** 3 key themes emerged:
1. Babies were extremely ill at hospital admission, with extended stays in intensive care required.
2. Identification of diabetes was not 'standardised' and perceived a 'chance' finding.
3. Adding glucose to NBS was universally considered extremely positive.

**Conclusions** Diagnosis of neonatal diabetes is frequently delayed, resulting in critically ill presentation with prolonged intensive care support, additional healthcare costs and familial distress. Potential to detect hyperglycaemia earlier was universally endorsed by parents with no negative consequences identified. Although further study including a larger number of individuals is needed to confirm our findings this study provides the first evidence of acceptability of glucose testing fulfilling Wilson-Jungner criteria for implementation within the NBS programme.

## INTRODUCTION

Neonatal diabetes mellitus (NDM) presenting less than 6 months of age is the result of severe insulin deficiency, leading

### Strengths and limitations of this study

► This study gained in-depth parental insight into the identification of neonatal diabetes and acceptability of newborn glucose screening.
► Families were from a variety of ethnic backgrounds and affected by five different genetic causes of neonatal diabetes, with varied treatments and prognosis.
► Patient and public involvement was central and findings were shared with participants to ensure trustworthiness and credibility of data and analysis.
► Limitations were the small number of families interviewed and therefore findings may not represent the views of all families with neonatal diabetes.

to hyperglycaemia and has a high burden of morbidity and mortality. NDM has an incidence of approximately 1:100 000 but reports range from 1 in 25 000 to 1:500 000 live births.[1–4] There are 23 known genetic causes which account for >80% of cases of diabetes diagnosed less than 6 months of age. Most patients with NDM present with severe hyperglycaemia and diabetic ketoacidosis, associated with prolonged hospital admission. Adverse outcomes include lifelong neurological damage or death with children suffering quadriplegia and brain damage, as a result of diabetic ketoacidosis with cerebral oedema and coning, requiring lifelong institutional support.[5–7] This morbidity and mortality is likely to be a marked underestimate as brain damage and death from ketoacidosis is often unrecognised or unrecorded. The prognosis is linked to the severity of the hyperglycaemia and ketoacidosis and crucially, the speed with which the disease is recognised and treatment implemented. The median duration of diabetes at the time of genetic testing for monogenic causes of NDM decreased from more than 4 years before 2005 to less than 3

months after 2012.[8] This was due to the introduction of a very simple patient selection and testing strategy; rapid, high-throughput robotic Sanger sequencing analysis of common causes of NDM genes by the Exeter laboratory offered free of charge for any patient in the world with diabetes diagnosed under 6 months. Widespread education and increased recognition of neonatal diabetes as a rare type of diabetes in addition to easily accessible free genetic testing contributed to the significant reduction in time from diabetes diagnosis to a confirmed molecular genetic diagnosis.

A correct genetic diagnosis of neonatal diabetes is important as it determines the most effective form of treatment. For the most common subtypes of neonatal diabetes due to *KCNJ11* or *ABCC8* mutations most individuals (>90%) can be transferred from insulin injections to sulfonylurea tablets with marked improvement in glycaemic control, with reductions in hypoglycaemia and blood glucose monitoring.[9–12] Twenty per cent of mutations in these genes cause neurological impairment and improvements in motor and cognitive function have been reported, with the greatest benefits seen with earlier sulfonylurea treatment, leading to improved glucose control, neurological function and quality of life for the patient and their families.[7 13–15]

In the UK, all neonates are screened at day 5 of life as part of standard routine postnatal care. Sample collection involves a heel prick blood sample collected on specialist bloodspot paper cards. These are analysed by one of a network of National Health Service (NHS) Neonatal Screening Laboratories for nine rare but serious metabolic diseases. This NHS routine screening service offers the potential for early detection of hyperglycaemia, as insulin deficiency from birth results in hyperglycaemia that can be detected in the first few days of life,[16] thereby facilitating earlier treatment and genetic diagnosis of NDM.

For a specific test to be considered viable for a screening programme, a number of criteria have been suggested.[17] The most widely used are those proposed by Wilson-Jungner, which are designed to appraise the validity of a screening programme and have been adapted for application to screening in the UK.[18] The original criteria are:
A. The condition being screened for should be an important health problem.
B. The natural history of the condition should be well understood.
C. There should be a detectable early stage.
D. Treatment at an early stage should be of more benefit than at a later stage.
E. A suitable test should be devised for the early stage.
F. The test should be acceptable.
G. Intervals for repeating the test should be determined.
H. Adequate health service provision should be made for the extra clinical workload resulting from screening.
I. The risks, both physical and psychological, should be less than the benefits.
J. The costs should be balanced against the benefits.

Screening for NDM fulfils most criteria, with compelling evidence that early identification of NDM would avoid many of the severe complications and mortality associated with early unrecognised presentation of diabetes.

However, the test should also be considered 'acceptable'.[18]

Our study aimed to gain understanding of parental experience of the presentation and recognition of neonatal diabetes and their views on the acceptability of adding glucose testing to the national newborn bloodspot screening (NBS) programme.

## METHODS

### Recruitment

Participants were recruited from the NEWBIE 1 study, which comprised families that had genetic testing for neonatal diabetes at the Royal Devon and Exeter NHS Foundation Trust Clinical Laboratories from 2013 to 2020. NEWBIE aimed to establish the diagnostic accuracy of newborn screening for neonatal diabetes by assessing glucose at day 5 of life and the clinicians of 139 children across the UK were contacted to ask permission to approach these families. Agreement was provided by 48 parents who were contacted sequentially, and verbal consent was taken by telephone (by KL) and consent forms were posted to potential participants, with 25 subsequently returning written consent. Twenty of 25 had a confirmed genetic diagnosis and therefore comprised the potential cohort for this qualitative study.

As the families had young children and were located across the UK, telephone interviews were considered both convenient and acceptable to parents and were arranged for a time and date to suit themient. Ten interviews were conducted as data saturation was achieved at this point and so no further families were contacted at this stage for this qualitative study.

A qualitative approach, from a social constructivist stance, was used to gain in-depth insight into the presentation and recognition of neonatal diabetes. A social constructivist perspective allows the experience of illness to be viewed from multiple perspectives as social and cultural constructs, with individuals' experiences of illness being inextricably linked with their experience of life. Parent's stories or narratives therefore provide a means of contextualising illness experience in a holistic biographical context. The focus of the interviews was on gaining an understanding of the meaning these events surrounding their child's diagnosis had for the families being studied, and their views on the acceptability of adding glucose to the NBS programme.

### Patient and public involvement

Patient and public involvement has been integral throughout the project. The study was initially discussed with 12 members (three with diabetes) of the National Institute for Health Research Exeter Clinical Research Facility user group and all agreed the project was feasible

**Table 1** Characteristics of the families taking part

| Participant | Gene affected | PNDM/TNDM* | Ethnic group | Parent affected (confirmed by genetic analysis) | Currently in remission |
|---|---|---|---|---|---|
| 1 | *KCNJ11* | PNDM | Western European | No | N/A |
| 2 | *6q24* | TNDM | Western European | No | Yes |
| 3 | *KCNJ11* | TNDM | Pakistani | Parents heterozygous (but unaffected, first cousins) | Yes |
| 4 | *ABCC8* | TNDM | Western European | Mother | Yes |
| 5 | *GCK* | PNDM | Arabic | Parents heterozygous | N/A |
| 6 | *6q24* | TNDM | Western European | No | Yes |
| 7 | *KCNJ11* | PNDM | Western European | Mother | N/A |
| 8 | *INS* | PNDM | Western European | Mother | N/A |
| 9 | *KCNJ11* | PNDM | Chinese | No | N/A |
| 10 | *KCNJ11* | TNDM | Western European | Mother | Yes |

N/A, not applicable; PNDM, Permanent Neonatal Diabetes Mellitus; TNDM, Transient Neonatal Diabetes Mellitus.

and would provide no inconvenience. Three parents of children affected by neonatal diabetes were also contacted and unanimously supported the proposal and two were actively involved in the creating and reviewing patient information. They will also assist us in producing summaries of the project for dissemination to ensure these are clear to non-healthcare professionals and can have maximum impact. The findings of the study have also been shared with the participants as described in the Discussion section.

In-depth telephone interviews were conducted with 10 parents (eight mothers and two fathers) of 14 children with neonatal diabetes, currently aged 1–14 years, median 2 years. Eight of the children had transient neonatal diabetes: *KCNJ11* (n=5), *ABCC8* (n=1), *6q24* (n=2) all of whom were currently in remission and six had permanent neonatal diabetes: *KCNJ11* (n=4), *INS* (n=1), homozygous *GCK* (n=1). Families were from a variety of ethnic groups: Western European (n=7), Arabic (n=1), Pakistani (n=1), Chinese (n=1) (table 1).

### Data collection and analysis

Telephone interviews were conducted with all participants by an experienced qualitative researcher (MS), who introduced herself and her role within this study. She had previously been in contact with four of the parents in her capacity as senior nurse within the monogenic diabetes team when providing clinical advice regarding the management of the neonatal diabetes following a genetic diagnosis. Duration of the interviews was determined by the interviewee's responses, but typically lasted 45–60 min. Parents were previously asked to identify and remain in a private, quiet location for the duration of the interview. The interviews were semistructured with open questions, using a qualitative interview guide which focused on six domains:
1. Experiences of pregnancy/postdelivery period.
2. Recognition of diabetes.
3. Experience of genetic diagnosis.
4. Response to diagnosis and experiences since.
5. Awareness of NBS tests.
6. Views of adding glucose to the NBS and anything else they felt was important to allow issues that were significant to the families to emerge.

The questions asked and prompts given around these domains were deliberately broad and open ended to allow the participants to describe the issues they considered most important, for example, '*I'm interested to hear about your response to the diagnosis of neonatal diabetes and your experiences since....*' (see online supplemental data). The approach taken was flexible and iterative.

The interviews were audio recorded to ensure accurate data collection and the interviewer also made notes. Interviews were subsequently transcribed verbatim, although names and identifying features were abbreviated to preserve anonymity. All participants were assigned a unique study ID. A thematic content analysis was undertaken using an inductive process, this was conducted manually due to the small number of interviews conducted. Transcripts were read and re-read to enable the researchers to become immersed in the data. Highlighting and colour coding of sections of text was used to identify potential issues and areas of interest. Notes were made on the transcripts to indicate the possible topic areas and these were compared across the different transcripts to generate a list of common themes which were considered and refined into themes of key importance. Interviews were concluded after 10 interviews as data saturation was achieved with no new themes emerging from the data. The transcripts were independently analysed by MS and BAK. The initial data analysis was revisited to refine the themes and develop subthemes with participant quotes identified which supported the themes. Themes were discussed with the research team and discrepancies resolved by consensus. Recurring patterns were identified

and relationships between the themes within and across transcripts were recorded. Anonymised interview transcripts were stored on a password-protected computer within a secure (swipe card access only) research environment. The results reported in this paper focus specifically on the issues relating to the period prior to/during the diabetes diagnosis and subsequent genetic test and the parents' views of NBS for glucose, other issues will be described as appropriate in a separate manuscript.

## RESULTS

Three key themes and eight subthemes emerged:

### Theme 1: babies were extremely ill when admitted to hospital

#### Subtheme A: parent's concerns not being taken seriously resulted in delays in admission

Although symptoms of hyperglycaemia had been reported by parents prior to admission, these had been dismissed by other healthcare professionals and led to delays in diagnosis.

> They had no suspicion at all that something can be wrong….even GP, doctors look at her 'Oh yeah she's so good, she's fine' … One day before (hospital admission) I said 'She's drinking a lot of milk …and she's sick a lot now' and (they said) 'Oh it can happen it's reflux' (but) …she was drinking and drinking and drinking. (Participant 1)

Polydipsia is a recognised sign of hyperglycaemia in children and adults diagnosed with diabetes but that was not recognised in this case.

Other causes of symptoms were assumed and delays in admission resulted.

> She'd not grown since she was 6 weeks old and she kept throwing up her bottles and always demanding more, she was always grumpy ….so we took her to the doctors but there was a sickness bug going round and they said 'Bring her back on Monday.' (Participant 4)

Vomiting is also known to be a feature of diabetic ketoacidosis but again was dismissed as having an alternative cause in both these cases.

#### Subtheme B: delays in admission resulted in long stays in intensive care required

Many of the babies were considered to be extremely ill when they were finally admitted to hospital and families recognised healthcare professionals' concerns and babies were frequently taken straight to the neonatal intensive care units (ICU).

> They were very worried, she was taken direct to ICU… it was a nightmare for me. (Participant 5)

Both families and healthcare professionals' concern about the well-being of the baby was clearly evident. The uncertainty of the situation added to the families' concerns.

Consequently, long stays of several weeks in intensive care were common and were likely to have resulted from delays in diagnosis and degree of hyperglycaemia on admission.

> He was in hospital for 5 weeks altogether. (Participant 6)

> He was in 'neonatal' (ICU) for about 3 weeks. (Participant 7)

#### Subtheme C: the severity of the babies' condition on admission increased parental anxiety

There was clear concern regarding the baby's chances of survival and this impacted emotionally on the families.

> Stressed, depressed, don't know what was going to happen to him, all a worry… it was frightening and scary and horrible. (Participant 6)

Delays in recognising the deteriorating health of the neonates and the significance of the symptoms resulted in deferral in admission, increased distress for the parents and prolonged hospital stays.

### Theme 2: identification of diabetes was not 'standardised' and often perceived to be a 'chance' finding

#### Subtheme D: testing for glucose was often attributed to a single healthcare professional

A number of families considered testing of blood glucose was down to 'luck' or the attention of an individual healthcare professional as opposed to a routinely performed test.

> One of the nurses, she was not asked to do so but, she did a urine sample and they found her sugars were abnormal and that is what made her go into the hospital again so it was lucky for H to find it so early. (Participant 5)

Participants considered testing for glucose to be random, rather than a systematic investigation prior to ICU admission.

Where an individual healthcare professional was regarded as being the 'single person' who had identified the problem, questions were raised about what might have transpired if that healthcare professional was not involved.

> It was just this one good doctor who came …and he was marvellous and luckily he come or I don't know what would have happened …I mean would B have been in a worse state? …maybe he'd have 'gone' for ever. (Participant 6)

Parents were clearly aware their babies could have died if the diagnosis had been further postponed.

#### Subtheme E: testing for glucose was advised by the families themselves

One family, where the baby's mother was known to have diabetes herself, had insisted their baby's blood glucose tested, despite being advised to the contrary.

She had an abscess above her bottom …and she was admitted … (my) Dad said 'Mum's diabetic, can you test her (the baby's) blood sugars please?' and they tried to fob us off they were 'No, no, no, it won't be anything like that' but he was adamant …and that's when they did and they (the blood sugars) were 18 … she was 2 weeks (of age)… it could have completely got missed. (Participant 8)

This reinforces the importance of healthcare professionals recognising and responding to families' concerns.

### Subtheme F: glucose values on testing were substantially raised

Extremely high levels of glucose on admission were common, suggesting the diagnosis of diabetes had clearly been delayed.

It was 30s, 36 nearly touching 40 (mmol/L) sometimes. (Participant 6)

The blood sugar was extremely high, around 40–50 (mmol/L). (Participant 9)

### Subtheme G: healthcare professionals did not anticipate diabetes in neonates

Healthcare professionals were not perceived to expect diabetes to be present at such an early age and parents highlighted the need for education.

They obviously didn't have a clue about this different type of diabetes …they kept saying his blood sugars were high and they didn't really expect anything like this …they weren't aware … I do really think they need to know more. (Participant 7)

Healthcare professional's lack of familiarity with neonatal diabetes added to the anxieties felt by the families.

The most difficult part for me was the beginning, because literally nobody knew what was going on, it wasn't common, nobody knew what to do and it really scared us. (Participant 2)

The period prior to diagnosis and the eventual recognition of a rare condition which many healthcare professionals would not be familiar with increased parental anxiety.

### Subtheme H: despite perceiving glucose testing to be a chance finding parents were positive about the care their child received

Despite a lack of experience with neonatal diabetes, parents praised the care given by the healthcare professionals and recognised this was also a difficult circumstance for the teams involved.

Even though all the medical staff were absolutely amazing I can't fault them …this was a very unknown situation for them… I could see nobody knew what to do really. (Participant 2)

Due to the rarity of neonatal diabetes it was unsurprising that many healthcare professionals were unfamiliar with

this condition, however parents were hugely appreciative of the care their child received and this influenced one family's view of the NHS.

I have been very, very grateful for everything that the diabetes specialist has done for our baby, it actually changed the way I think about the UK health system. (Participant 9)

Recognition of diabetes and testing of blood glucose was considered by parents to be predominantly a consequence of the location of their baby, for example, in an intensive care setting or down to individual healthcare professionals, as opposed to a routine test undertaken in unwell neonates. Despite healthcare professionals' unfamiliarity with neonatal diabetes the families were very grateful for the care their children received.

### Theme 3: adding glucose to the NBS test was universally considered extremely positive

Every single parent interviewed was positive about the possibility of adding a glucose test onto the NBS programme.

I think it's an absolutely fantastic idea I really, really do it's brilliant … and I think if it wasn't for me carrying the gene …they wouldn't probably have got tested, so I think it's an absolutely fantastic idea. (Participant 10)

Parents could not see any disadvantages to NBS as this would ensure blood glucose was measured close after birth and would prevent missed or delayed diagnosis.

They recognised this could avoid symptoms of hyperglycaemia being missed.

I think it's a good idea, that would have been ideal because we went through weeks of … the symptoms of diabetes but not picking up on it. (Participant 4)

And recognised that testing at day 5 of life could prevent deterioration in the condition of babies with diabetes.

That would be a great idea, I mean before it develops even further… discovering this at a very early stage is going to be extremely helpful. (Participant 9)

Potential detection at an early stage was considered very valuable. Parents considered this approach would avoid potential problems from a delayed diagnosis.

She was lucky to be diagnosed at 10 days… I wonder how she survived. In the future … it would be much better if they were diagnosed earlier and treated earlier, rather than leaving for later days and leading to damage for the child. (Participant 5)

They highlighted the possibility of preventing babies becoming seriously ill with all the fear that caused.

My baby went into coma and nearly died …so if they find it at day 5 …we could have avoided that. (Participant 1)

Parents recognised that NBS for glucose could prevent babies from becoming seriously unwell and dying.

Parents were unanimous in the view that adding glucose to the NBS programme was a positive step which would reduce delayed diagnosis and deterioration in the health of babies with diabetes and prevent other families from going through the trauma they had as a consequence of a delayed diagnosis.

## DISCUSSION

Our study provides the first evidence of parental support for the addition of glucose to the newborn screening programme and in-depth insights into the experiences around diabetes diagnosis during the neonatal period in five different genetic subtypes. The key themes identified in this study highlight the risk of a delayed diagnosis with hyperglycaemic symptoms often missed and diabetes diagnosis often considered by families to be a chance finding. This led to babies being extremely ill when the diagnosis was finally made, resulting in long hospital stays and high levels of distress among the families. These delays influenced the families' perceptions of newborn glucose screening as important to prevent other families going through the same situation with the potential for a much earlier diagnosis.

Recognition and diagnosis of neonatal diabetes is frequently delayed with babies presenting with extreme hyperglycaemia, requiring extended intensive care support. This leads to high NHS care costs and parental distress. Parents considered the addition of a glucose measurement to NBS would reduce delays in the diagnosis of neonatal diabetes and prevent the extent of the suffering they experienced with the deteriorating health of their baby. No parents indicated any concerns or suggested any potential negative consequences of NBS screening for glucose within the interviews.

We have previously shown that glucose levels at day 5 of life in those with neonatal diabetes are already raised and markedly higher than the normal range,[16] therefore the introduction of glucose as part of the newborn screening programme could prevent delays in recognition and ensure prompt treatment of neonatal diabetes.

Previous studies have explored attitudes to expanding newborn screening in other conditions but many have focused on the views of clinicians[19] or parents with 'healthy' children[20] with far less attention paid to the views of families living with the conditions.[21] We considered it of paramount importance to gain understanding of those actually experiencing the impact of neonatal diabetes to inform NBS for glucose as it is those who have lived through the experience who are best placed to appreciate the issues involved,

Parental views are recognised to be influenced by contextual factors such as personal experience.[21] In families with haemophilia the vast majority (77%) supported newborn screening predominantly because they considered it a means to facilitate early support and treatment,

inform decisions about future pregnancies and prevent the 'diagnostic odyssey' and difficulties associated with a later diagnosis.[22] Concerns about the detrimental impact, in terms of bonding and parental stress, that an unsought and serious diagnosis can have on early parent-child relationships have been highlighted in cystic fibrosis[23 24] and spinal muscular atrophy (SMA).[25] Despite the majority (70%) of families and individuals living with SMA being in favour of newborn genetic screening due to delays in diagnosis, those not in favour (whose babies had an extremely curtailed lifespan or a longer period of time before the onset of symptoms) were concerned that newborn genetic screening would prevent families from enjoying carefree time with their baby before symptoms emerged.[25]

### Strengths and limitations

A key strength of this study is that participants were from a variety of backgrounds and represented five different genetic causes of neonatal diabetes, with varied treatments and prognosis. Parents were all able to provide 'insider information' on their personal experiences of a delayed diagnosis of neonatal diabetes in their families. Copies of the draft paper were sent to the participants to ensure the trustworthiness and credibility of the data and analysis, with an invitation to provide any feedback to the research team. Only four of the participants responded to the draft paper and they indicated they were satisfied with the data and information included and none requested any edits. The number of families interviewed was small and therefore our findings may not represent the views of all families with neonatal diabetes, although data saturation was reached and clear themes identified. However, further study including more individuals would be required to support our findings.

### CONCLUSION

Our study contributes to an emerging literature that considers the acceptability of newborn screening practices from the vantage point of parents with experience of a child with the condition. Although further study including a larger number of individuals is needed to confirm our findings this study provides new evidence that adding glucose to the NBS programme is not only acceptable to families but endorsed by them, providing the first confirmation of acceptability of testing, fulfilling one of the Wilson-Jungner criteria for adding glucose as part of the NBS programme.

**Acknowledgements** The NIHR Exeter Clinical Research Facility is a partnership between the University of Exeter Medical School College of Medicine and Health, and Royal Devon and Exeter NHS Foundation Trust. This project is supported by the National Institute for Health Research (NIHR) Exeter Clinical Research Facility.

**Contributors** MS, BAK and TJM conceptualised the study and developed the methodology. MS, BAK and KL recruited the participants. MS conducted and transcribed the interviews. MS and BAK analysed the data. MS prepared the

original draft. BAK, KL and TJM reviewed and edited the drafts and MS revised the subsequent editions.

**Funding** MS is a National Institute for Health Research (NIHR) 70@70 Senior Nurse and Midwife Research Leader.

**Disclaimer** The views expressed are those of the author(s) and not necessarily those of the NIHR or the Department of Health and Social Care.

**Competing interests** None declared.

**Patient consent for publication** Not required.

**Ethics approval** Ethical approval was granted by the South West-Cornwall and Plymouth Research Ethics Committee (NREC number: 18/SW/0070).

**Provenance and peer review** Not commissioned; externally peer reviewed.

**Data availability statement** Data are available upon reasonable request. Requests for access to data, and to discuss potential collaborations, should be made in writing in the first instance to the Chief Investigator Professor Timothy McDonald, timothy.mcdonald@nhs.net.

**ORCID iD**
Maggie Shepherd http://orcid.org/0000-0003-2660-0955

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
