## [Reviewer comments · BMJ Open]

ARTICLE DETAILS

TITLE (PROVISIONAL)	Parental experiences of a diagnosis of neonatal diabetes and perceptions of new-born screening for glucose: A qualitative study
AUTHORS	Shepherd, Maggie; Knight, Bridget; Laskey, Katherine; McDonald, Timothy

VERSION 1 – REVIEW

REVIEWER	Xiuzhen Li Guangzhou Women and Children's Medical Center, China
REVIEW RETURNED	13-Feb-2020

GENERAL COMMENTS	This study is interesting and meaningful for suggesting the involvement of 'glucose' in new-born screening. Comments: 1) Short title - It need to be shortened. 2) Abstract - The results should be more concise as the words of the participants could be removed. - The strengths and limitations of this study need to be shortened. 3) Introduction - In L90, "In Exeter, the National and International referral and diagnostic centre for neonatal diabetes, we are aware of at least 9 cases of children suffering quadriplegia and brain damage as a result of diabetic ketoacidosis with cerebral oedema and coning requiring lifelong institutional support [5-7]". In Ref 6, only pancreatic tissue of this patient was investigated. I wonder if this reference fits better with the last sentence "Most patients with NDM present with severe hyperglycaemia and diabetic ketoacidosis, associated with prolonged hospital admission and adverse outcomes including lifelong neurological damage or death [4,5]". 4) Method - In Table 1, the expansion of PNDM and TNDM should be given. - In Table 1, how to define the "Parents affected" column, by genetic analysis or blood glucose testing or both? It should be addressed. - More details of the thematic content analysis should be given to understand how it works; for example, the inductive, manual process used to identify key themes and codes. 5) Results - More contents should be extracted from the participants' words. For example, in L215, "Babies were extremely ill when admitted to hospital, with long stays in intensive care required". Detailed data of the patients' conditions should be shown to indicate the severity of the illness, such as "blood glucose level", "if diabetic ketoacidosis or not", "length from the appearance of symptoms to
--

	confirmed diagnosis”, “length of hospitalization”, “length in ICU”, and so on. 6) Discussion - It would be better to give more descriptions around the three themes of this study to emphasize the risk of delayed diagnosis, and provide further support for the addition of glucose to new-born screening. 7) Conclusion - To make the conclusion more concise and clear, some contents could be transferred to discussion; for example, “Recognition and diagnosis of neonatal diabetes is frequently delayed with babies presenting with extreme hyperglycaemia, requiring extended intensive care support. This leads to high NHS care costs and parental distress.” and “Parents considered addition of a glucose measurement to NBS would reduce delays in the diagnosis of neonatal diabetes and prevent the extent of the suffering they experienced with the deteriorating health of their baby.” could be further discussed.
--	--

REVIEWER	abdelhadi habeb Prince Mohamed Bin Abdulaziz Hospital, Madinah, KSA
REVIEW RETURNED	04-Mar-2020

GENERAL COMMENTS	this is an interesting study looking into the parental perception of adding NDM to the newborn screening program. Although the interviews were meticulously conducted the number of participants is too small to provide convincing evidence for "public acceptance" for the proposed screening program based on the J&W criteria or the UK national screening committee. in order to fulfill the "acceptance" criteria" a large number of parents with healthy and NDM children, general public as well as healthcare professionals should be surveyed. this point has to be highlighted by the authors in the discussion and conclusions. I have the following comments to improve the manuscript: 1- Abstract: there is no need to mention table 1 or quotations of what parents said in the result and may be useful in the conclusions to acknowledge that further study including a large number is needed to confirm our findings etc 2- introduction: a) No need for subtitles in the introduction. b) the incidence of NDM varies between countries from 1:21,000 in KSA to 1:500,000 in Austria but the figure of 1;90000 is just for Italy. Maybe there is a stronger argument for adding the NDM to newborn screening program in areas where is the incidence is higher Results: would it possible to put the important parental quotations in a box? Discussion: a) the authors focused a lot on the benefits of adding the NDM to the screening program but it is useful to address the pros and con. b) better to move the the paragraph of we previous shown after the first paragraph c) 10 parents is a small number so the the limitation should be addressed more (see above) d) please explain what you mean of data saturation were reached
---

REVIEWER	Dr R Lotto Liverpool John Moores University
REVIEW RETURNED	18-May-2020

GENERAL COMMENTS	This is well written manuscript, and I really enjoyed reading it. Thank you
--

	There are a few minor issues that I feel would improve the article  1. This is perhaps a personal preference, but it is unusual to see quotations in an abstract. A more nuanced explanation of each theme would be provide the reader with an interpretation of the theme, rather than being left to interpret a quote themselves. 2. There are a number of places in the introduction/background that the reader is left hanging. Line 96 - why was there a large drop between 2005 and 2012. Was something implemented? Make this clear. Line 104 - previously talk about 28 types, but then name 2. I appreciate that you state these are the most common, but taken as written, naming them is meaningless without further information. Expand or cut. 3. Exter's experience can be cut/combined (to free up more words) as you discuss this further down, using other relevant literature. 4. Line 126-127 not entirely clear what you are saying. Can you rephrase. 5. 197 Audio record not tape record 6. 204 - Just state the facts - data was looked at by x and y. Currently sounds like the second reviewer was an add on. 7. 289-290 This could be expanded upon. Patient perception that no-one knew what was going on. Is there anything in the clinician literature to draw on. 8. Discussion - really starts to pull the results together, but ends rather abruptly. I was left dying to read more! Please expand.
--	--

REVIEWER	Alexa Craig, MD Maine Medical Center USA
REVIEW RETURNED	20-May-2020

GENERAL COMMENTS	Title: Parental experiences of a diagnosis of neonatal diabetes and perceptions of new-born screening for glucose: A qualitative study Authors: Maggie H Shepherd, Bridget A Knight, Katherine Laskey, Timothy J McDonald Summary: The authors of this paper present the results of a qualitative study on parental perceptions of adding a glucose test to the newborn screen to identify neonates with diabetes earlier. Ten parents (8 mothers and 2 fathers) of 14 infants were interviewed and the transcribed interviews were analyzed using a thematic content analysis methodology. Three overarching themes emerged; 1. At presentation, neonates were critically ill, 2. Discovery of neonatal diabetes was perceived to be a chance finding and 3. Sentiments toward early testing of glucose were positive. The authors conclude that this essential step of fulfilling Wilson-Jungner criteria for implementation of glucose testing within the NBS has been fulfilled while acknowledging the limitation that their sample size is small. Critiques to the researchers: Abstract: I would be careful with adverb use such as immense cost and extreme family distress. Much of the conclusion is restating what is known about this problem rather than what this paper adds in terms of knowledge. Introduction: It may be helpful to include in your introduction what the normal range of neonatal blood glucose is as measured in the UK as a reference for when parents are describing abnormal values in the quotations. For readers from outside the UK, the assay is entirely different (US normal is around 60 for instance). Would also be helpful to explain more about the Wilson-Jungner criteria. Methods: The authors describe that a subset of parents were invited to take part in the interview. How many parents were evaluated for participation and how was the decision made to offer
--

	participation to a subset? This must be clearly explained to avoid the impression that the authors may have been biased toward certain families. I think it may help to show a diagram or write out that X# were considered for involvement, the contact was made by the nurse for X# and then X# returned the signed consent form. You could make a table showing that the group that did not respond were not different from the group that did in terms of certain important characteristics. The open-ended questions, since there were only 6 of them, should be listed in a table or provided as a supplement. Why did the research team do this manually and not use software to perform the analysis? How was the coding done? Mention is made of subthemes, yet these are not discussed in the results. Copies of the manuscript were sent to families to “validate” but no data is provided about whether or not they had any edits to make etc. How do you know they even read the manuscript? Was a phone call made to ask them after the fact what they thought?? Results: I do not think that the quotes provided in theme #1 all support that theme. There are quotes about the parents’ concerns not being taken seriously which seem separate from the theme of “babies were extremely ill at presentation and had long hospital stays”. Are there more quotes from other families to support the long ICU stay? Perhaps a table showing more examples? Theme#2-“chance findings of NDM”. Again in this section, there seems to be a mix of themes. How does parental appreciation of care provided fit into the diagnosis was made by chance?? It seems like it might be a separate theme. Theme #3: this is well presented-all quotes seems to directly support the theme. In the list of six domains that the authors said in the methods they were going to address, no data is presented for experience of pregnancy, the specific experience of getting a genetic diagnosis or the response to that diagnosis or awareness of NBS. I think this data should be included or explanation given as to why this was not addressed (e.g. will be described in a separate manuscript??). Table 1: If possible, provide explanations for abbreviations in table (e.g. PNDM and TNDM). Discussion: The discussion does a nice job of linking parental perceptions of NBS in other diseases, but it doesn’t really put the findings from this study in any context of existing literature. For example, the authors state that other literature has focused on clinician experience or that of parents of healthy children and has not assessed families with the disorder. Why in this study did the authors only interview affected families? It seems like getting more perspectives would have been more informative. I think this should be acknowledged as a significant weakness in the study and the very small sample size needs to also be more emphasized.
--	---

VERSION 1 – AUTHOR RESPONSE

Reviewer: 1

Reviewer Name: Xiuzhen Li

Institution and Country: Guangzhou Women and Children’s Medical Center, China Please state any competing interests or state ‘None declared’: None declared.

Please leave your comments for the authors below This study is interesting and meaningful for suggesting the involvement of ‘glucose’ in new-born screening.

Comments:

1) Short title

- It need to be shortened.

This has been shortened to: Parent's perception of new-born glucose screening: A qualitative study

2) Abstract

- The results should be more concise as the words of the participants could be removed.

Participant quotes have been removed

- The strengths and limitations of this study need to be shortened.

This section has been reduced

3) Introduction

- In L90, "In Exeter, the National and International referral and diagnostic centre for neonatal diabetes, we are aware of at least 9 cases of children suffering quadriplegia and brain damage as a result of diabetic ketoacidosis with cerebral oedema and coning requiring lifelong institutional support [5-7]". In Ref 6, only pancreatic tissue of this patient was investigated. I wonder if this reference fits better with the last sentence "Most patients with NDM present with severe hyperglycaemia and diabetic ketoacidosis, associated with prolonged hospital admission and adverse outcomes including lifelong neurological damage or death [4,5]".

Thank you - we have amended this

4) Method

- In Table 1, the expansion of PNDM and TNDM should be given.

This detail has been added below the table

- In Table 1, how to define the "Parents affected" column, by genetic analysis or blood glucose testing or both? It should be addressed.

We have added in the table that the parent being affected was confirmed by genetic analysis

- More details of the thematic content analysis should be given to understand how it works; for example, the inductive, manual process used to identify key themes and codes.

Thank you this additional detail has now been included

5) Results

- More contents should be extracted from the participants' words. For example, in L215, "Babies were extremely ill when admitted to hospital, with long stays in intensive care required". Detailed data of the patients' conditions should be shown to indicate the severity of the illness, such as "blood glucose level", "if diabetic ketoacidosis or not", "length from the appearance of symptoms to confirmed diagnosis", "length of hospitalization", "length in ICU", and so on.

As the children were diagnosed at different hospitals across the UK and only referred to the Royal Devon and Exeter Hospital for genetic testing we do not have access to details regarding if they had diabetic ketoacidosis, length of appearance of symptoms to confirmed diagnosis, length of hospitalisation or length of stay in ICU apart from the information provided by the parents which has been included in their quotes

6) Discussion

- It would be better to give more descriptions around the three themes of this study to emphasize the risk of delayed diagnosis, and provide further support for the addition of glucose to new-born screening.

Thank you – more descriptions have now been included in the discussion

7) Conclusion

- To make the conclusion more concise and clear, some contents could be transferred to discussion; for example, "Recognition and diagnosis of neonatal diabetes is frequently delayed with babies presenting with extreme hyperglycaemia, requiring extended intensive care support. This leads to high NHS care costs and parental distress."

This has been moved to the discussion

and "Parents considered addition of a glucose measurement to NBS would reduce delays in the diagnosis of neonatal diabetes and prevent the extent of the suffering they experienced with the deteriorating health of their baby." could be further discussed.

This sentence has also been moved to the discussion and further discussion added

Reviewer: 2

Reviewer Name: abdelhadi habeb

Institution and Country: Prince Mohamed Bin Abdulaziz Hospital, Madinah, KSA Please state any competing interests or state 'None declared': none

Please leave your comments for the authors below this is an interesting study looking into the parental perception of adding NDM to the newborn screening program. Although the interviews were meticulously conducted the number of participants is too small to provide convincing evidence for "public acceptance" for the proposed screening program based on the J&W criteria or the UK national screening committee. in order to fulfill the "acceptance" criteria" a large number of parents with healthy and NDM children, general public as well as healthcare professionals should be surveyed. this point has to be highlighted by the authors in the discussion and conclusions.

Thank you- we know that the test for NBS is acceptable (heel prick) as over 800,000 parents take up the test every year in the UK- the question posed to us by the National Screening Committee was- is it acceptable for this condition (neonatal diabetes) ? The National screening committee consider that it is important to find out if it is acceptable to parents of affected children. We appreciate that the views of a larger number of parents with affected children would be helpful and we've added a statement to that effect to our limitations and conclusion. We have included more information about the J&W criteria and added an additional 2 references to support this.

I have the following comments to improve the manuscript:

1- Abstract: there is no need to mention table 1 or quotations of what parents said in the result Participant quotes have been removed

and may be useful in the conclusions to acknowledge that further study including a large number is needed to confirm our findings etc

We now acknowledge in the conclusions that further study including a larger number of individuals is needed to confirm our findings

2- introduction: a) No need for subtitles in the introduction.

We have removed the subtitles in the introduction

b) the incidence of NDM varies between countries from 1:21,000 in KSA to 1:500,000 in Austria but the figure of 1;90000 is just for Italy. Maybe there is a stronger argument for adding the NDM to newborn screening program in areas where is the incidence is higher

Thank you – we have now amended this to read 'NDM has an incidence of approximately 1:100,000 but reports range from between 1 in 25,000 to 1:500,000 live births'. We don't actually consider the incidence to be different but detection rates may vary. We have added this additional reference: Grulich-Henn J, Wagner V, Thon A, et al. Entities and frequency of neonatal diabetes: data from the diabetes documentation and quality management system (DPV). Diabet Med 2010; 27:709.

Results: would it possible to put the important parental quotations in a box?

If the Editor thinks this would be appropriate for the journal we could move some of the parental quotes out of the text and into a box but we would suggest that unless the quotes in the box were repeated in the text this would affect the flow of the article

Discussion: a) the authors focused a lot on the benefits of adding the NDM to the screening program but it is useful to address the pros and con.

We have now addressed this in the manuscript by including the following sentence 'No parents indicated any concerns or suggested any potential negative consequences of NBS screening for glucose within the interviews.

b) better to move the the paragraph of we previous shown after the first paragraph

Thank you – we have moved this paragraph

c) 10 parents is a small number so the the limitation should be addressed more (see above)

We have added a sentence to our limitations paragraph and also to the conclusion to this effect

d) please explain what you mean of data saturation were reached

This is already described towards the end of the methods section so we have not repeated the explanation in the discussion

Reviewer: 3

Reviewer Name: Dr R Lotto

Institution and Country: Liverpool John Moores University Please state any competing interests or state 'None declared': None

Please leave your comments for the authors below This is well written manuscript, and I really enjoyed reading it. Thank you There are a few minor issues that I feel would improve the article 1. This is perhaps a personal preference, but it is unusual to see quotations in an abstract.

Participant quotes have been removed

A more nuanced explanation of each theme would be provide the reader with an interpretation of the theme, rather than being left to interpret a quote themselves.

Thank you - we have now provided additional interpretation after the quotes

2. There are a number of places in the introduction/background that the reader is left hanging. Line 96 - why was there a large drop between 2005 and 2012. Was something implemented? Make this clear. Thank you – we have now added the following sentences to clarify: 'This was due to the introduction of a very simple patient selection and testing strategy; rapid, high-throughput robotic Sanger sequencing analysis of common cause NDM genes by the Exeter laboratory offered free of charge for any patient in the world with diabetes diagnosed under 6 months. Wide spread education and increased recognition of neonatal diabetes as a rare type of diabetes in addition to easily accessible free genetic testing contributed to the significant reduction in time from diabetes diagnosis to a confirmed molecular genetic diagnosis.

Line 104 - previously talk about 28 types, but then name 2. I appreciate that you state these are the most common, but taken as written, naming them is meaningless without further information. Expand or cut.

KCNJ11 and ABCC8 neonatal diabetes account for around 50% of all cases and have treatment implications (in contrast to the other genetic causes). We thought it important to highlight that there are a large number of different genetic causes of neonatal diabetes and that the two most common causes can be optimally treated with sulphonylureas as opposed to insulin so have left both pieces of information in the text (as this was not highlighted as an issue by the other reviewers) but can alter if the Editor prefers.

3. Exter's experience can be cut/combined (to free up more words) as you discuss this further down, using other relevant literature.

Thank you - we have combined this as advised

4. Line 126-127 not entirely clear what you are saying. Can you rephrase.

Thank you - we have provided the following detail: 'A social constructivist perspective allows the experience of illness to be viewed from multiple perspectives as social and cultural constructs, with individuals experiences of illness being inextricably linked with their experience of life. Parent's stories or narratives therefore provide a means of contextualising illness experience in a holistic biographical context'.

5. 197 Audio record not tape record
This has been changed

6. 204 - Just state the facts - data was looked at by x and y. Currently sounds like the second reviewer was an add on.
This has been amended as suggested

7. 289-290 This could be expanded upon. Patient perception that no-one knew what was going on. Is there anything in the clinician literature to draw on.
We have expanded on the explanation following this quote to highlight that due to the rarity of neonatal diabetes it was unsurprising that many healthcare professionals very unfamiliar with this condition

8. Discussion - really starts to pull the results together, but ends rather abruptly. I was left dying to read more! Please expand.
Thank you – we have now revised the discussion and added more to this section

Reviewer: 4

Reviewer Name: Alexa Craig, MD

Institution and Country:

Maine Medical Center

USA

Please state any competing interests or state 'None declared': None declared

Please leave your comments for the authors below

Title: Parental experiences of a diagnosis of neonatal diabetes and perceptions of new-born screening for glucose: A qualitative study

Authors: Maggie H Shepherd, Bridget A Knight, Katherine Laskey, Timothy J McDonald

Summary: The authors of this paper present the results of a qualitative study on parental perceptions of adding a glucose test to the newborn screen to identify neonates with diabetes earlier. Ten parents (8 mothers and 2 fathers) of 14 infants were interviewed and the transcribed interviews were analyzed using a thematic content analysis methodology. Three overarching themes emerged; 1. At presentation, neonates were critically ill, 2. Discovery of neonatal diabetes was perceived to be a chance finding and 3. Sentiments toward early testing of glucose were positive. The authors conclude that this essential step of fulfilling Wilson-Jungner criteria for implementation of glucose testing within the NBS has been fulfilled while acknowledging the limitation that their sample size is small.

Critiques to the researchers:

Abstract: I would be careful with adverb use such as immense cost and extreme family distress.

Thank you – we have changed 'immense' to 'additional' and removed 'extreme'

Much of the conclusion is restating what is known about this problem rather than what this paper adds in terms of knowledge.

We have re-worded the conclusion

Introduction: It may be helpful to include in your introduction what the normal range of neonatal blood glucose is as measured in the UK as a reference for when parents are describing abnormal values in the quotations. For readers from outside the UK, the assay is entirely different (US normal is around 60 for instance).

I'm afraid it is not possible to include a normal range of neonatal blood glucose as it varies with age and whether breast or formula fed. Therefore one alternative would be to offer the US units in mg/dL as well as mmol/L for non-UK readers if the editors believe this is appropriate. We have however added 'mmol/L' where the blood glucose levels have been described by the parents so the readers

are aware of the units referred to

Would also be helpful to explain more about the Wilson-Jungner criteria.

Thank you – we have now included additional information in the text to explain more about the Wilson-Jungner criteria and added an additional 2 references to support this

Methods: The authors describe that a subset of parents were invited to take part in the interview. How many parents were evaluated for participation and how was the decision made to offer participation to a subset? This must be clearly explained to avoid the impression that the authors may have been biased toward certain families. I think it may help to show a diagram or write out that X# were considered for involvement, the contact was made by the nurse for X# and then X# returned the signed consent form. You could make a table showing that the group that did not respond were not different from the group that did in terms of certain important characteristics.

Thank you we have now added this information-

Participants were recruited from the NEWBIE 1 study, which comprised families that had genetic testing for neonatal diabetes at the Royal Devon and Exeter NHS Foundation Trust Clinical Laboratories from 2013 – 2020. Ethical approval was granted by the South West – Cornwall & Plymouth Research Ethics Committee - NREC no: 18/SW/0070). NEWBIE aimed to establish the diagnostic accuracy of new-born screening for neonatal diabetes by assessing glucose at day five of life and the clinicians of 139 children across the UK were contacted to ask permission to approach these families. Agreement was provided by 48 parents who were contacted sequentially and verbal consent was taken by telephone (by KL) and consent forms were posted to potential participants, with 25 subsequently returning written consent. 20/25 had a confirmed genetic diagnosis and therefore comprised the potential cohort for this qualitative study.

As the families had young children and were located across the UK, telephone interviews were considered both convenient and acceptable to parents and were arranged for a time and date that was convenient. 10 interviews were conducted as data saturation was achieved at this point and so no further families were contacted at this stage for this qualitative study.

The open-ended questions, since there were only 6 of them, should be listed in a table or provided as a supplement.

Thank you - we have now provided this information as supplementary.

Why did the research team do this manually and not use software to perform the analysis? How was the coding done?

Thank you- this has now been explained in more detail in the text

Mention is made of subthemes, yet these are not discussed in the results.

Thank you – we have now added details of the subthemes within the manuscript

Copies of the manuscript were sent to families to “validate” but no data is provided about whether or not they had any edits to make etc. How do you know they even read the manuscript? Was a phone call made to ask them after the fact what they thought??

Thank you – we have now added the following sentence ‘Only four of the participants responded to the draft paper and they indicated they were satisfied with the data and information included and none requested any edits.’

Results: I do not think that the quotes provided in theme #1 all support that theme. There are quotes about the parents’ concerns not being taken seriously which seem ‘separate from the theme of “babies were extremely ill at presentation and had long hospital stays”. Are there more quotes from

other families to support the long ICU stay? Perhaps a table showing more examples?

Thank you – we have now included details of the sub-themes and separated this section in the elements within each sub-theme. There are more quotes from other families indicating long ICU stays were common but we have not included these as they are very similar to the quotes already included e.g. providing details of how many weeks their babies stayed in ICU. It is not usual to provide multiple examples of similar issues and so we have not included more examples relating to the long ICU stay but can confirm this was a common finding

Theme#2-“chance findings of NDM”. Again in this section, there seems to be a mix of themes. How does parental appreciation of care provided fit into the diagnosis was made by chance?? It seems like it might be a separate theme.

Thank you – again we have now included the details of the sub-themes within this section which we hope provides the reader with a clearer picture of how the sub-themes relate to the main theme.

Theme #3: this is well presented-all quotes seems to directly support the theme.

In the list of six domains that the authors said in the methods they were going to address, no data is presented for experience of pregnancy, the specific experience of getting a genetic diagnosis or the response to that diagnosis or awareness of NBS. I think this data should be included or explanation given as to why this was not addressed (e.g. will be described in a separate manuscript??).

Thank you We have added the following sentence in the text to clarify why these areas are not included: ‘The results reported in this paper focus specifically on the issues relating to the period prior to / during the diabetes diagnosis and subsequent genetic test and the parents views of NBS for glucose, other issues will be described as appropriate in a separate manuscript’.

We felt this was important to focus on these areas to ensure clarity of message. The experience of receiving a genetic diagnosis and response to the diagnosis has previously been studied in families with KCNJ11 and ABCC8 neonatal diabetes, the experiences of pregnancy were straightforward in these cases and all families were aware of NBS for other conditions at day 5 of life and these topics were therefore were not considered by the authors to add to this particular manuscript.

Table 1: If possible, provide explanations for abbreviations in table (e.g. PNDM and TNDM).

This detail has been added below the table

Discussion: The discussion does a nice job of linking parental perceptions of NBS in other diseases, but it doesn't really put the findings from this study in any context of existing literature. For example, the authors state that other literature has focused on clinician experience or that of parents of healthy children and has not assessed families with the disorder. Why in this study did the authors only interview affected families? It seems like getting more perspectives would have been more informative. I think this should be acknowledged as a significant weakness in the study and the very small sample size needs to also be more emphasized.

Thank you - we have made revisions to the discussion. However the actual test (heel prick) has already been established as acceptable- now the question is, is it acceptable to detect this condition (diabetes) early by adding a glucose test to the NBS programme ? Therefore it is the families that count and this is why we have chosen to interview affected families only. If we were proposing introducing heel pricks for the first time we would need a more general group including unaffected families

VERSION 2 – REVIEW

REVIEWER	Xiuzhen Li Guangzhou Women and Children's Medical Center,China
REVIEW RETURNED	21-Jul-2020

GENERAL COMMENTS	Revised manuscript has improved and is suitable to be published.
--

REVIEWER	Dr Robyn Lotto Liverpool John Moores University UK
REVIEW RETURNED	10-Jul-2020

GENERAL COMMENTS	Thank you for addressing the comments provided in the review. I have no further comments.
---

REVIEWER	Alexa Craig, MD Maine Medical Center, USA
REVIEW RETURNED	08-Jul-2020

GENERAL COMMENTS	I think the authors did a really fantastic job with their edits. I think the paper is much clearer now and I appreciate the increased number of quotations that really help show their sentiments.
--